# Handgrip strength cutoff value predicting successful extubation in mechanically ventilated patients

**Narongkorn Saiphoklang*, Nattawadee Mokkongphai**

Division of Pulmonary and Critical Care Medicine, Department of Internal Medicine, Faculty of Medicine, Thammasat University, Pathum Thani, Thailand

* m_narongkorn@hotmail.com

**Data Availability Statement:** All relevant data are within the manuscript and its Supporting information files.

## Abstract

### Background

Handgrip strength (HGS) is an alternative tool to evaluate respiratory muscle function. HGS cutoff value indicating extubation success or failure has not been investigated. This study aimed to determine HGS cutoff value to predict successful extubation.

### Methods

A prospective study was conducted. Patients requiring intubated mechanical ventilation with intubation $\geq$ 48 hours in medical wards were recruited. HGS test was performed at 10 minutes before and 30 minutes after spontaneous breathing trial (SBT). Rapid shallow breathing index (RSBI) was measured at 10 minutes before SBT.

### Results

Ninety-three patients (58% men) were included. Mean age was 71.6 ± 15.2 years. Weaning failure rate was 6.5%. The area under the ROC curve of 0.84 for the best HGS cutoff value at 10 minutes before SBT was 12.7 kg, with 75.9% sensitivity and 83.3% specificity (P = 0.005). The best HSG cutoff value at 30 minutes after SBT was 14.9 kg, with the area under the ROC curve of 0.82, with 58.6% sensitivity and 83.3% specificity (P = 0.009). The best RSBI cutoff value was 43.5 breaths/min/L, with the area under the ROC curve of 0.46, 33.3% sensitivity and 66.6% specificity (P = 0.737).

### Conclusions

HGS may be a predictive tool to guide extubation with better sensitivity and specificity than RSBI. A prospective study is needed to verify HGS test as adjunctive to RSBI in ventilator weaning protocol.

**Funding:** The work was supported by Faculty of Medicine, Thammasat University, Thailand. The funders had no role in study design, data collection and analysis, decision to publish, or preparation of the manuscript.

**Competing interests:** The authors have declared that no competing interests exist.

## Introduction

Respiratory failure is the main illness requiring mechanical ventilation (MV) [1]. Evaluation of readiness for weaning from MV is an essential process for reducing morbidity and mortality. Successful weaning can reduce complications of prolonged ventilation while failed weaning can result in reintubation. Assessment for weaning in mechanically ventilated patients can be divided into clinical assessment and physical assessment. Clinical assessments include adequacy of cough reflex, reduction of sputum production and lack of indications requiring intubation. Physical assessments are vital signs and clinical stability, adequate oxygenation, adequate lung function, and rapid shallow breathing index (RSBI) less than 105 breaths/minute/liter [1–4]. Common causes of weaning failure are increasing respiratory load or cardiac load, neuromuscular conditions, neuropsychiatric disorders, metabolic derangements, nutritional problems, and anemia [1]. The transition from full ventilatory support to spontaneous breathing trial (SBT) requires adequate respiratory muscle strength to maintain breathing and acceptable gas exchange [5].

RSBI is one of the most common methods used for weaning process because it is the most accurate predictor of failure in weaning patients from MV [6]. This measure is easy to conduct using pressure support ventilation or spirometer. Nevertheless, some patients will require reintubation and institution of MV despite meeting established weaning criteria, while some patients not meeting the criteria can be successfully weaned from MV.

The results of several studies revealed that weaning and extubation should be guided by several parameters, and not only by respiratory ones [7]. There are many tools other than RSBI to predict success or failure in the weaning process such as handgrip strength (HSG), heart rate variability, sleep quality, diaphragmatic dysfunction, and oxidative stress markers [2]. A previous study showed that HSG can be used for adjunctive monitoring along with maximum inspiratory pressure for weaning from prolonged MV at a long-term acute-care hospital [8]. A study of ventilator weaning using simple motor tasks, including hand grasping and tongue protrusion, showed that the inability to follow simple motor commands was a predictor of extubation failure in critically ill neurological patients [9]. HGS could not predict extubation failure but it could predict difficult weaning in mechanically ventilated patients [10]. Another previous study showed the relationship between respiratory muscle strength and HGS in the healthy elderly, as well as the maximal inspiratory pressure and maximal expiratory pressure significantly associated with HSG [11].

This study aimed to determine the HGS cutoff value to predict successful extubation in mechanically ventilated patients.

## Methods

### Study design and participants

A prospective study was conducted from January 2018 to January 2020 in two 30-bed general medical wards and a 10-bed medical intensive care unit (ICU) at Thammasat University Hospital, an 800-bed tertiary care teaching hospital in the northern Bangkok conurbation, Thailand. Patients requiring MV with tracheal tube intubation for at least 48 hours and aged at least 18 years were recruited. All recruits were able to cooperate fully, and able to do the HGS test, assessed by the ability to follow at least two simple commands (i.e., raise right arm and do a 'thumb up' sign). Exclusion criteria were death before weaning from MV, less than 48 hours of MV, transfer to other hospitals, self-extubation, accidental extubation, re-intubation before enrollment, undergoing tracheostomy, inability to perform HGS test, and treatment with vasopressor/inotropic drugs. The 34 patients successfully recruited for this study

were also among the patients who participated at the same time in our previously published study [12].

Weaning was conducted according to the standards of the European and American respiratory/intensive care societies [1]. Weaning was attempted as early as possible during the patients' illnesses with a two-step approach in which readiness for weaning was assessed daily according to the standard criteria of the European and American respiratory/intensive care societies [1]. Patients who fulfilled these criteria underwent a SBT. The duration of SBT was 30–120 minutes and consisted of either breathing with a T-piece or a weaning trial undergoing 5–8 cmH$_2$O pressure support with 5 cm H$_2$O positive end-expiratory pressure. When patients successfully passed the SBT, the physician in charge, in collaboration with the attending medical staff initiated the weaning process.

If a patient failed the initial SBT, MV was reinstituted and the physician reviewed the possible reversible causes of the weaning failure, including respiratory factors (e.g. bronchospasm, secretion obstruction, pulmonary edema); cardiovascular factors (e.g. congestive heart failure); psychoneurologic factors (e.g. delirium); metabolic factors (e.g., electrolyte imbalances, dysglycemia); nutritional factors (e.g., malnutrition and anemia). The SBT was repeated the following day if the patient then appeared ready to wean.

A patient was rated as successfully weaned when he or she was extubated and breathing spontaneously without any invasive or noninvasive ventilatory support for ≥ 48 hours. Concordantly, weaning failure was defined as either the failure of SBT or the need for reintubation within 48 hours following extubation [1].

Extubation failure in this study was defined as inability to sustain spontaneous breathing after removal of an endotracheal tube and need for reintubation or non-invasive ventilation within 48 hours.

Participants were classified into 3 weaning groups: (1) simple weaning: successful weaning and extubation on the first attempt without difficulty, (2) difficult weaning: failure of initial weaning and the need for up to three SBTs for as many as 7 days from the first SBT to achieve successful weaning; and (3) prolonged weaning: failure of 3 or more weaning attempts or the need for longer than 7 days of weaning after the first SBT [1].

Demographics and baseline characteristics were collected for all participants. During SBT, data on MV weaning was collected including vital signs, RSBI, type of SBT, time of SBT success, and time of extubation. HGS was tested at 48 hours after intubation, and 10 minutes and 30 minutes after SBT. RSBI was performed at 10 minutes before SBT. The physician in charge did not know HGS results.

Ethic approval was obtained from the Human Ethics Committee of Thammasat University (IRB No. MTU-EC-IM-0-197/60). All participants provided written informed consent.

## Measurements

Investigator explained the HGS measurement process to patients and had them perform one measurement to ensure correct methods. The first attempt was not recorded. Maximal grip strength from 3 subsequent efforts for each hand was recorded using a specialized dynamometer (Jamar; Asimow Engineering Co; Santa Monica, CA, USA). The measurements were made at rest with the hand unsupported, and with the elbow at 90º flexion, and with wrist in neutral position.

## Statistical analysis

Categorical data was shown as number (%). Continuous data was shown as mean ± standard deviation. The optimal HSG cutoff value was determined using the Receiver Operator

Characteristic (ROC) curve. A two-sided p-value < 0.05 was considered statistically significant. Statistical analyses were performed using SPSS version 20.0 software (IBM Corp., Armonk, NY, USA).

## Results

One hundred twenty mechanically ventilated patients were recruited and 93 of these were included in the final analysis (see Fig 1).

Men were 58%. Mean age was 71.6 ± 15.2 years. Most patients (84.9%) were admitted in general medical wards. APACHE II score was 13.5 ± 4.7. Most patients were intubated from pneumonia (39.8%). The most common type of SBT was pressure support ventilation (74.2%). Weaning success was 6.5%. Incidence of simple, difficult, and prolonged weaning were 77.4%, 20.4% and 2.2%, respectively (see Table 1).

The weaning success group had significantly higher HGS than the weaning failure group over time. There was no significant difference in RSBI between the weaning success and the weaning failure groups (see Table 2).

The causes for extubation failure were pneumonia (33.3%), pulmonary edema (16.7%), marked airway secretions (16.7%), bronchospasm (16.7%) and hypoalbuminemia (16.7%).

The area under the ROC curve of 0.84 (95% CI; 0.67–1.00, P = 0.005) for the best cutoff value of HGS at 10 minutes before SBT was 12.7 kg, with 75.9% sensitivity and 83.3% specificity. The best cutoff value of HSG at 30 minutes after SBT was 14.9 kg, which showed the area under the ROC curve of 0.82 (95% CI; 0.64–1.00, P = 0.009). The best RSBI cutoff value was 43.5 breaths/min/L, with the area under the ROC curve of 0.46 (95% CI; 0.16–0.75, P = 0.737), 33.3% sensitivity and 66.6% specificity (see Fig 2 and Table 3).

There were no statistically significant differences in HGS and RSBI between simple, difficult, and prolonged weaning groups (see S1 Table).

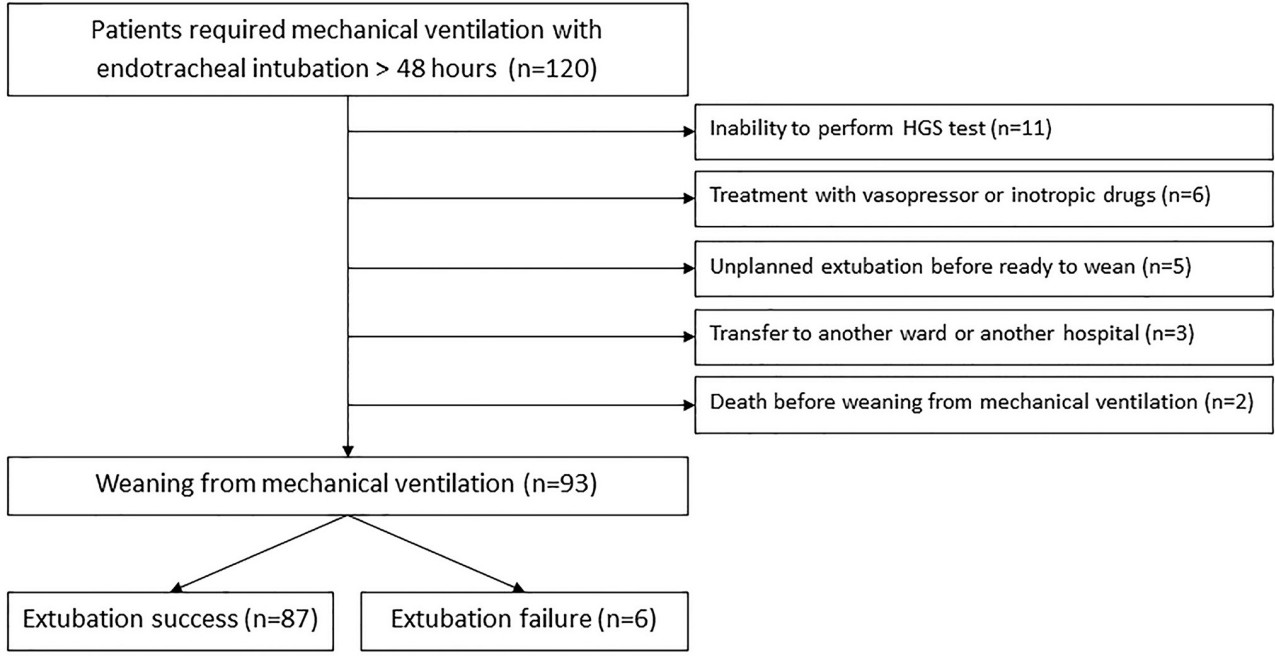

**Fig 1. Study flowchart indicates inclusion and exclusion population.**

**Table 1. Baseline characteristics of the patients.**

| Characteristic | N = 93 |
|---|---|
| Age, years | 71.6 ± 15.2 |
| Male | 54 (58.1) |
| Body mass index, kg/m$^2$ | 23.5 ± 4.6 |
| Dominant right hand | 87 (93.5) |
| **Ward** | |
| General medical wards | 79 (84.9) |
| Medical intensive care unit | 14 (15.1) |
| **Underlying disease** | |
| Hypertension | 39 (41.9) |
| Diabetes | 23 (24.7) |
| COPD | 15 (16.1) |
| Malignancy | 13 (14.0) |
| Atrial fibrillation | 10 (10.8) |
| Chronic kidney disease | 8 (8.6) |
| Chronic heart failure | 7 (7.5) |
| APACHE II score, points | 13.5 ± 4.7 |
| **Indication for intubation with mechanical ventilation** | |
| Pneumonia | 37 (39.8) |
| Airway protection | 12 (12.9) |
| AECOPD | 11 (11.8) |
| Congestive heart failure | 9 (9.7) |
| Volume overload | 6 (6.5) |
| Septic shock | 2 (2.2) |
| Atrial fibrillation | 1 (1.1) |
| Others | 15 (16.0) |
| **Mode of ventilator weaning** | |
| Pressure support ventilation | 69 (74.2) |
| T-piece | 24 (25.8) |
| SBT duration, minutes | 113.7 ± 20.4 |
| **Weaning outcome** | |
| Weaning success | 87 (93.5) |
| Weaning failure | 6 (6.5) |
| **Weaning group** | |
| Simple | 72 (77.4) |
| Difficult | 19 (20.4) |
| Prolonged | 2 (2.2) |

Data are presented as n (%), mean±SD.

AECOPD = acute exacerbation of COPD, APACHE II = acute physiology and chronic health evaluation II,

COPD = chronic obstructive pulmonary disease, SBT = spontaneous breathing trial.

## Discussion

This is the first prospective study to determines the HGS cutoff value for predicting successful weaning from mechanical ventilation. This measure is a simple and easy method similar to RSBI but it depends on patient cooperation. Interestingly, HGS had higher sensitivity and specificity than RSBI. This study suggests that the HGS cutoff value may be applied in elderly

**Table 2. Comparison of patient characteristics, handgrip strength and rapid shallow breathing index between weaning success and weaning failure.**

| Variable | Total | Weaning success | Weaning failure | P-value |
|---|---|---|---|---|
| | N = 93 | n = 87 | n = 6 | |
| Age, years | 71.6 ± 15.2 | 72.3 ± 14.3 | 61.3 ± 24.9 | 0.332 |
| Male | 54 (58.1) | 51 (58.6) | 3 (50.0) | 0.693 |
| Body mass index, kg/m$^2$ | 23.5 ± 4.6 | 23.4 ± 4.4 | 24.1 ± 6.9 | 0.726 |
| General medical wards | 79 (84.9) | 76 (87.4) | 3 (50.0) | 0.042 |
| Hypertension | 39 (41.9) | 37 (42.5) | 2 (33.3) | 1.000 |
| Diabetes | 23 (24.7) | 21 (24.1) | 2 (33.3) | 0.635 |
| COPD | 15 (16.1) | 15 (17.2) | 0 (0) | 0.585 |
| Malignancy | 13 (14.0) | 13 (14.9) | 0 (0) | 0.590 |
| Pneumonia | 37 (39.8) | 36 (41.4) | 1 (16.7) | 0.397 |
| Airway protection | 12 (12.9) | 12 (13.8) | 0 (0) | 1.000 |
| APACHE II score, points | 13.5 ± 4.7 | 13.3 ± 4.61 | 16.3 ± 5.2 | 0.125 |
| Pressure support ventilation on weaning | 69 (74.2) | 67 (77.0) | 2 (33.3) | 0.037 |
| SBT duration, minutes | 113.7 ± 20.4 | 113.3 ± 21.1 | 120.0 ± 0.0 | 0.437 |
| HGS at 48 hours after intubation, kg | 15.6 ± 7.0 | 16.2 ± 6.8 | 7.3 ± 4.7 | 0.002 |
| HGS at 10 minutes before SBT, kg | 15.8 ± 6.7 | 16.3 ± 6.5 | 8.2 ± 5.3 | 0.004 |
| HGS at 30 minutes before SBT, kg | 15.9 ± 6.6 | 16.4 ± 6.3 | 8.6 ± 5.6 | 0.004 |
| RSBI at 10 minutes before SBT, breaths/min/L | 40.2 ± 9.2 | 40.0 ± 8.9 | 42.2 ± 14.4 | 0.590 |

Data are presented as n (%), mean±SD.

APACHE II = acute physiology and chronic health evaluation II, COPD = chronic obstructive pulmonary disease, HGS = handgrip strength, RSBI = rapid shallow breathing index, SBT = spontaneous breathing trial.

populations because our participants were elder (mean age of 71.6 years) and it may be more suitable for patients with restrictive lung diseases who do not meet RSBI criteria but can be successfully weaned from MV. Majority of our patients (85%) were admitted in general medical wards due to the limitation of ICU beds in our hospital. However, our medical staffs in general medical wards were well trained in respiratory care and intensive care for patients requiring MV and there were sufficient devices for respiratory and noninvasive hemodynamic monitoring in these wards.

RSBI was the most common predictor in 15 previous studies, followed by age and the maximum inspiratory pressure in 7 of the studies reviewed [7]. Baptistella AR, et al suggests that weaning and extubation should be guided by several parameters, and not only respiratory ones [7]. RSBI of 105 breaths/minute/liter or less indicated weaning success with 78% positive predictive value and 95% negative predictive value according to a study of Yang KL and Tobin MJ [6]. However, some patients will require reintubation and institution of MV despite meeting established weaning criteria, while some patients not meeting the criteria can be successfully liberated from the ventilator. Our study found that the best RSBI cutoff value (43.5 breaths/min/L) is lower than in the previous study [6]. It may have resulted from different patient setting; some of our patients with COPD (16%) might breathe with high lung volumes and low respiratory rates following improvement of their acute illness.

HGS shows a significant correlation with respiratory muscle strength assessed by maximal inspiratory pressure in the healthy young and middle-age subjects [13] and in the healthy elderlies [11]. It is one of several tools to predict success or failure in the weaning process [2] and it can predict difficult weaning in mechanically ventilated patients according to a study by Cottereau G, et al. [10]. Saiphoklang N, et al found that HGS may be a predictive tool for

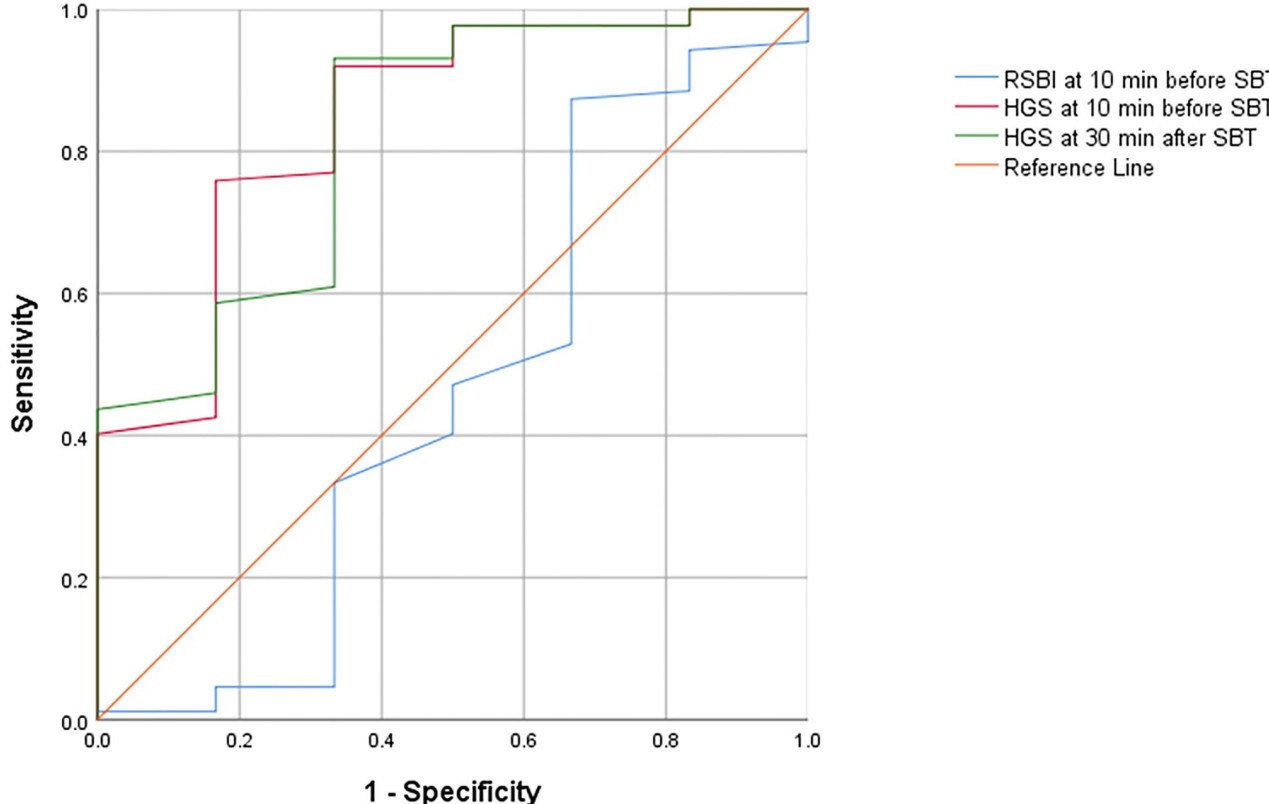

**Fig 2. The receiver operating characteristic (ROC) plot of handgrip strength (HGS) and rapid shallow breathing index (RSBI).** The area under the ROC curve of HGS at 10 min before SBT is 0.842 (95% CI: 0.67–1.00), HSG at 30 min after SBT is 0.822 (95% CI: 0.64–1.00), and RSBI is 43.5 (95% CI: 0.16–0.75). Blue line = RSBI at 10 min before SBT, Red line = HGS at 10 min before SBT, Green line = HGS at 30 min after SBT, Orange line = reference line.

extubation failure in patients requiring MV. Low strength was associated with increased re-intubation rate [12]. Moreover, Ali NA, et al demonstrated that HGS was associated with poor clinical outcomes and increased hospital mortality in patients with ICU-acquired paresis (ICUAP) [14]. Furthermore, from Verceles AC, et al. [15], it can be an assessment tool of a physical therapy program for patients with ICUAP in long-term acute care at hospital. As well, a study of Mohamed-Hussein AAR, et al showed that HGS may be a good predictor for MV duration, extubation outcome, and ICU mortality [16].

Our study demonstrates that both HGS at 10 minutes before SBT and at 30 minutes after SBT can predict weaning success. These findings may also suggest an indirect association

**Table 3. Cutoff values of handgrip strength and rapid shallow breathing index to distinguish between weaning success and weaning failure.**

| Variable | Cutoff value | AUC | 95% CI | Sensitivity (%) | Specificity (%) | PPV (%) | NPV (%) | P-value |
|---|---|---|---|---|---|---|---|---|
| HGS at 10 minutes before SBT, kg | 12.7 | 0.842 | 0.67–1.00 | 75.9 | 83.3 | 52.2 | 48.5 | 0.005 |
| HGS at 30 minutes after SBT, kg | 14.9 | 0.822 | 0.64–1.00 | 58.6 | 83.3 | 45.7 | 55.0 | 0.009 |
| RSBI, breaths/min/L | 43.5 | 0.459 | 0.16–0.75 | 33.3 | 66.6 | 37.5 | 63.6 | 0.737 |

AUC = area under the ROC curve, CI = confidence interval, HGS = handgrip strength, NPV = negative predictive values, PPV = positive predictive values, RSBI = rapid shallow breathing index, SBT = spontaneous breathing trial.

between HSG and diaphragmatic muscle strength either before or during SBT. Such association is expected to remain even after patients become exhausted and have reduced grip strength after SBT. Therefore, HGS might be applied as a predictive tool for weaning from MV like these circumstances. However, assessment of HGS may be limited by the patients' level of cooperation and hand deformity.

This study has a few limitations. The first is that this study cannot demonstrate correlation between RSBI and HGS to predict successful extubation in mechanically ventilated patients. The second, we cannot analyse for subgroup cutoff values (e.g., age, gender) due to the small size of the study population. The small sample size could have explained the wide confidence interval in the results, despite the point estimates indicating good accuracy, thus the results should be cautiously interpreted. A third limitation is that this study included only patients from medical wards, thus may not apply to surgical patients. A future study may determine correlation between RSBI and HGS, as well as implement HSG to a ventilator liberation protocol.

## Conclusion

HGS was able to be a predictive tool to guide extubation with better sensitivity and specificity than RSBI. Prolonged weaning and weaning failure had low incidence in our setting. A prospective study is needed to verify HGS test in ventilator weaning protocol and as an adjunctive to RSBI.

## Supporting information

**S1 Table. Comparison of baseline characteristics, mechanical ventilation data, rapid shallow breathing index, and handgrip strength between 3 weaning groups.**
(DOCX)

## Acknowledgments

The authors would like to thank Michael Jan Everts and Dr Kanon Jatuworapruk, Faculty of Medicine, Thammasat University, for proofreading this manuscript.

## Author Contributions

**Conceptualization:** Narongkorn Saiphoklang, Nattawadee Mokkongphai.

**Data curation:** Narongkorn Saiphoklang, Nattawadee Mokkongphai.

**Formal analysis:** Narongkorn Saiphoklang.

**Funding acquisition:** Nattawadee Mokkongphai.

**Investigation:** Narongkorn Saiphoklang, Nattawadee Mokkongphai.

**Methodology:** Narongkorn Saiphoklang, Nattawadee Mokkongphai.

**Project administration:** Narongkorn Saiphoklang.

**Resources:** Narongkorn Saiphoklang, Nattawadee Mokkongphai.

**Supervision:** Narongkorn Saiphoklang.

**Validation:** Nattawadee Mokkongphai.

**Visualization:** Narongkorn Saiphoklang, Nattawadee Mokkongphai.

**Writing – original draft:** Narongkorn Saiphoklang, Nattawadee Mokkongphai.

**Writing – review & editing:** Narongkorn Saiphoklang, Nattawadee Mokkongphai.

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
