## [Decision Letter · Decision Letter 0]

27 May 2021

PONE-D-21-10826

Handgrip Strength Cutoff Value Predicting Successful Extubation in Mechanically Ventilated Patients

PLOS ONE

Dear Dr. Saiphoklang,

Thank you for submitting your manuscript to PLOS ONE. After careful consideration, we feel that it has merit but does not fully meet PLOS ONE’s publication criteria as it currently stands. Therefore, we invite you to submit a revised version of the manuscript that addresses the points raised during the review process.

This is certainly an interesting study and one that potentially makes a meaningful contribution to the literature in this area. However, I agree with the reviewers that there are several critical points that need to be addressed prior to consideration for publication. In addition to reviewer comments, please address the following concerns in your revision:

-the statistical analysis description is not sufficiently detailed for readers to understand what was done and why

-my biggest concern is that it is not clear how repeated weaning attempts were handled from an analysis standpoint. the flow chart shows ~93% success overall, but not all of these were on the first attempt--how were the patients/values for those that failed a first (or 2nd) attempt but ultimately extubated handled? was each weaning trial considered an independent event? there is likely within-patient correlation on grip strength so i am not certain that this is the best approach. ultimately, there are so few failures to extubate that it is difficult to draw firm conclusions from these results, and excessive sub-group analysis probably only clouds the picture, but i do think that an analysis of the first weaning trial only would be worthwhile. however, clearly stating what was done is of critical importance

-please clarify when and how these patients were consented as this potentially speaks to concerns over selection bias

We look forward to receiving your revised manuscript.

Kind regards,

Robert Ehrman, MD, MS

Academic Editor

PLOS ONE

Journal Requirements:

"The work was supported by Faculty of Medicine, Thammasat University, Thailand."

Reviewers' comments:

Reviewer's Responses to Questions

**Comments to the Author**

1. Is the manuscript technically sound, and do the data support the conclusions?

Reviewer #1: Yes

Reviewer #2: Yes

2. Has the statistical analysis been performed appropriately and rigorously? 

Reviewer #1: Yes

Reviewer #2: No

3. Have the authors made all data underlying the findings in their manuscript fully available?

Reviewer #1: Yes

Reviewer #2: Yes

4. Is the manuscript presented in an intelligible fashion and written in standard English?

Reviewer #1: Yes

Reviewer #2: Yes

5. Review Comments to the Author

Reviewer #1: This study aims to investigate the usefulness of handgrip strength (HGS) in the predicting successful extubation. They found HGS could be a useful predictive toll to guide extubation. Overall, the study is well-designed and the manuscript is well-written. I just have several suggestions.

1. I just wonder why more than 80% of patients were in general medical wards.

2. Please briefly describe study sites, including both general ward and ICU.

3. Please add something more in the table 2, including patients’ characteristics, use of sedation, neuromuscular blockage and steroid.

4. We need more discussion about HSG, not only the description of each study’s finding about HSG.

Reviewer #2: This is a very interesting article with a simple method to evaluate the patient’s readiness to be weaned/extubated.

Although, some important points need to be elucidate and some additional analysis should be performed.

- SBT failure and extubation failure should be considered different outcomes.

- The variation in the SBT duration (30 to 120 minutes) can interfere in the outcome, and even in the HGS result. For some patients, 120 min is a big challenge. The standardization of the SBT duration is very important (all patients with 30 min or all with 120 min). Another possibility is to analyze separated those with 30 min form those with 120 min.

- The type of SBT performed can affect the results. This method should be standardized or analyze separated patients test in PSV and those tested in t-piece.

- How many measures of the HGS was performed? Do you consider the first try? There was a training time or explanation?

- How did you define the patient’s capacity to perform the HGS? Glasgow coma scale? RASS? The capacity to follow 2 o 3 commands?

- How can you explain/discuss the low ROC value of RSBI in this cohort? This is very different from others studies that used RSBI.

6. PLOS authors have the option to publish the peer review history of their article (what does this mean?). If published, this will include your full peer review and any attached files.

Reviewer #1: No

Reviewer #2: **Yes: **Antuani Rafael Baptistella

---

## [Author Response · Author response to Decision Letter 0]

25 Jun 2021

Dear Reviewers of PLOS ONE:

I submitted an original contribution to PLOS ONE entitled “Handgrip Strength Cutoff Value Predicting Successful Extubation in Mechanically Ventilated Patients” (Manuscript Number PONE-D-21-10826). You have kindly expressed some interesting points with your comments. Therefore, my response to the comment is below.

RESPONSE TO REVIEWER 1:

REVIEWER COMMENT 1: I just wonder why more than 80% of patients were in general medical wards. 

RESPONSE: I have added more texts to explain the reasons in Paragraph 1 of Discussion section.

REVIEWER COMMENT 2: Please briefly describe study sites, including both general ward and ICU.

RESPONSE: I have added more descriptions in Paragraph 1 of Methods section (Study design and participants).

REVIEWER COMMENT 3: Please add something more in the table 2, including patients’ characteristics, use of sedation, neuromuscular blockage and steroid.

RESPONSE: I have added more data including patient characteristics in Table 2.

REVIEWER COMMENT 4: We need more discussion about HSG, not only the description of each study’s finding about HSG.

RESPONSE: I have added more discussions in Paragraph 1 to 4 of Discussion section.

RESPONSE TO REVIEWER 2:

REVIEWER COMMENT 1: SBT failure and extubation failure should be considered different outcomes.

RESPONSE: I have added data of extubation failure as weaning outcome in Table 1. REVIEWER COMMENT 2: The variation in the SBT duration (30 to 120 minutes) can interfere in the outcome, and even in the HGS result. For some patients, 120 min is a big challenge. The standardization of the SBT duration is very important (all patients with 30 min or all with 120 min). Another possibility is to analyze separated those with 30 min form those with 120 min.

RESPONSE: I have added more data on SBT duration in Table 1 and Table 2.

REVIEWER COMMENT 3: The type of SBT performed can affect the results. This method should be standardized or analyze separated patients test in PSV and those tested in t-piece.

RESPONSE: I have added PSV data in Table 2.

REVIEWER COMMENT 4: How many measures of the HGS was performed? Do you consider the first try? There was a training time or explanation?

RESPONSE: I have added more details in Methods section (Measurements).

REVIEWER COMMENT 5: How did you define the patient’s capacity to perform the HGS? Glasgow coma scale? RASS? The capacity to follow 2 o 3 commands?

RESPONSE: I have added more details in Paragraph 1 of Methods section (Study design and participants).

REVIEWER COMMENT 6: How can you explain/discuss the low ROC value of RSBI in this cohort? This is very different from others studies that used RSBI.

RESPONSE: I have added more discussions in Paragraph 2 of Discussion section.

I appreciate your consideration of my manuscript for publication in PLOS ONE.

Sincerely, 

Narongkorn Saiphoklang, M.D.

---

## [Decision Letter · Decision Letter 1]

21 Jul 2021

PONE-D-21-10826R1

Handgrip strength cutoff value predicting successful extubation in mechanically ventilated patients

PLOS ONE

Dear Dr. Saiphoklang,

Thank you for submitting your manuscript to PLOS ONE. After careful consideration, we feel that it has merit but does not fully meet PLOS ONE’s publication criteria as it currently stands. Therefore, we invite you to submit a revised version of the manuscript that addresses the points raised during the review process.

Thank you for the time and effort out forth in revising your manuscript. While it is substantially improved, a few issues remain that require clarification prior to proceeding with publication.

We look forward to receiving your revised manuscript.

Kind regards,

Robert Ehrman, MD, MS

Academic Editor

PLOS ONE

Journal Requirements:

Reviewers' comments:

Reviewer's Responses to Questions

**Comments to the Author**

1. If the authors have adequately addressed your comments raised in a previous round of review and you feel that this manuscript is now acceptable for publication, you may indicate that here to bypass the “Comments to the Author” section, enter your conflict of interest statement in the “Confidential to Editor” section, and submit your "Accept" recommendation.

Reviewer #1: All comments have been addressed

Reviewer #2: (No Response)

2. Is the manuscript technically sound, and do the data support the conclusions?

Reviewer #1: Yes

Reviewer #2: Partly

3. Has the statistical analysis been performed appropriately and rigorously? 

Reviewer #1: Yes

Reviewer #2: (No Response)

4. Have the authors made all data underlying the findings in their manuscript fully available?

Reviewer #1: Yes

Reviewer #2: (No Response)

5. Is the manuscript presented in an intelligible fashion and written in standard English?

Reviewer #1: Yes

Reviewer #2: (No Response)

6. Review Comments to the Author

Reviewer #1: The authors response well, so I have no more comment. Therefore, I recommend the present manuscript can be accepted.

Reviewer #2: Authors answered most of my comments, although, some important points are not completely clear.

1- The outcome used in table 2 is weaning success and failure, but the outcome described in methods and even mentioned in the title is extubation success or failure. Weaning success and extubation success are two different outcomes. Weaning success is defined as to pass for the SBT without any sign of intolerance, while extubation success is to breathe spontaneously for 48 hours without NIV dependence or reintubation. Authors must standardize the outcome used.

2- Regarding my previous comment “The type of SBT performed can affect the results. This method should be standardized or analyze separated patients test in PSV and those tested in t-piece.”

There was difference in the outcome in patients who performed the SBT in PSV or T piece. If you analyze the HGS in this 2 groups separately, the result would be the same?

3- Results section – first paragraph – “weaning success was 6.5%” – I suppose you mean “weaning failure”.

7. PLOS authors have the option to publish the peer review history of their article (what does this mean?). If published, this will include your full peer review and any attached files.

Reviewer #1: No

Reviewer #2: **Yes: **Antuani Rafael Baptistella

---

## [Author Response · Author response to Decision Letter 1]

3 Sep 2021

Dear Reviewers of PLOS ONE:

I submitted an original contribution to PLOS ONE entitled “Handgrip Strength Cutoff Value Predicting Successful Extubation in Mechanically Ventilated Patients” (Manuscript Number PONE-D-21-10826). You have kindly expressed some interesting points with your comments. Therefore, my response to the comment is below.

RESPONSE TO REVIEWER 1:

REVIEWER COMMENT: The authors response well, so I have no more comment. Therefore, I recommend the present manuscript can be accepted.

RESPONSE: I would like to gratefully thank the reviewer for kindly reviews. 

RESPONSE TO REVIEWER 2:

REVIEWER COMMENT 1: The outcome used in table 2 is weaning success and failure, but the outcome described in methods and even mentioned in the title is extubation success or failure. Weaning success and extubation success are two different outcomes. Weaning success is defined as to pass for the SBT without any sign of intolerance, while extubation success is to breathe spontaneously for 48 hours without NIV dependence or reintubation. Authors must standardize the outcome used.

RESPONSE: I have added the definition of extubation failure for this study in Method section, paragraph 5. Also, I have changed the weaning outcome to new words; extubation success and extubation failure, in Table 1, the heading title of Table 2, and Figure 1.

REVIEWER COMMENT 2: Regarding my previous comment “The type of SBT performed can affect the results. This method should be standardized or analyze separated patients test in PSV and those tested in t-piece.”

There was difference in the outcome in patients who performed the SBT in PSV or T piece. If you analyze the HGS in this 2 groups separately, the result would be the same?

RESPONSE: I have added T-piece data as the weaning mode in Table 2 which is separated from PSV data.

REVIEWER COMMENT 3: Results section – first paragraph – “weaning success was 6.5%” – I suppose you mean “weaning failure”.

RESPONSE: I have corrected this word to “extubation failure” in Results section. Also, I have corrected to new words in Abstract section, Results section, the heading title of Table 2 and 3, and Figure 1.

I appreciate your consideration of my manuscript for publication in PLOS ONE.

Sincerely, 

Narongkorn Saiphoklang, M.D.

---

## [Editor Report · Decision Letter 2]

6 Oct 2021

PONE-D-21-10826R2Handgrip strength cutoff value predicting successful extubation in mechanically ventilated patientsPLOS ONE

Dear Dr. Saiphoklang,

Thank you for submitting your manuscript to PLOS ONE. After careful consideration, we feel that it has merit but does not fully meet PLOS ONE’s publication criteria as it currently stands. Therefore, we invite you to submit a revised version of the manuscript that addresses the points raised during the review process.

Overall, the manuscript is improved in its current version. I would recommend adding a brief acknowledgement in the limitations section about the relatively small sample size--while the point estimates indicate good accuracy, the CIs are wide and thus the results should be cautiously interpreted.

We look forward to receiving your revised manuscript.

Kind regards,

Robert Ehrman, MD, MS

Academic Editor

PLOS ONE
---

## [Author Response · Author response to Decision Letter 2]

7 Oct 2021

RESPONSE TO EDITOR 

EDITOR COMMENT: Overall, the manuscript is improved in its current version. I would recommend adding a brief acknowledgement in the limitations section about the relatively small sample size--while the point estimates indicate good accuracy, the CIs are wide and thus the results should be cautiously interpreted.

RESPONSE: Thank you for your kindly comments. I have added this sentence in the limitations section.

I appreciate your consideration of my manuscript for publication in PLOS ONE.

Sincerely, 

Narongkorn Saiphoklang, M.D.

---

## [Editor Report · Decision Letter 3]

11 Oct 2021

Handgrip strength cutoff value predicting successful extubation in mechanically ventilated patients

PONE-D-21-10826R3

Dear Dr. Saiphoklang,

We’re pleased to inform you that your manuscript has been judged scientifically suitable for publication and will be formally accepted for publication once it meets all outstanding technical requirements.

Kind regards,

Robert Ehrman, MD, MS

Academic Editor

PLOS ONE
---

## [Editor Report · Acceptance letter]

13 Oct 2021

PONE-D-21-10826R3 

Handgrip strength cutoff value predicting successful extubation in mechanically ventilated patients 

Dear Dr. Saiphoklang:

I'm pleased to inform you that your manuscript has been deemed suitable for publication in PLOS ONE. Congratulations! Your manuscript is now with our production department. 

Kind regards, 

on behalf of

Dr. Robert Ehrman 

Academic Editor

PLOS ONE